



# Water level fluctuations drive bank instability in a hypertidal estuary

Andrea Gasparotto[1,2], Stephen E. Darby[2], Julian Leyland[2], Paul A. Carling[2]

[1]Department of Geography, College of Life and Environmental Sciences, University of Exeter, Exeter, EX4 4RJ, UK
[2]School of Geography and Environmental Science, University of Southampton, Southampton SO17 1BJ, UK

*Correspondence to*: Andrea Gasparotto (a.gasparotto@exeter.ac.uk)

**Abstract.** Hypertidal estuaries are very dynamic environments characterised by high tidal ranges (>6 m) that can experience rapid rates of bank retreat. Whilst a large body of work on the processes, rates, patterns and factors driving bank erosion has been undertaken in fluvial environments, the process mechanics affecting the stability of the banks with respect to mass failure

in hypertidal settings are not well documented. In this study, the processes and trends leading to bank failure and consequent retreat in hypertidal estuaries are treated within the context of the Severn Estuary (UK) by employing a combination of numerical models and field-based observations. Our results highlight that the periodic fluctuations in water level associated with the hypertidal environment drive regular fluctuations in the hydrostatic pressure exerted on the incipient failure surfaces that range from a confinement pressure of 0 kPa (at low tide) to ~100 kPa (at high tide). However, the relatively low

transmissivity of the fine-grained banks (that are typical of estuarine environments) results in low seepage inflow/outflow velocities (~$3 \times 10^{-10}$ m s$^{-1}$), such that variations in positive pore water pressures within the saturated bank are smaller, ranging between about 10 kPa (at low tide) to ~43 kPa (at high tides). This imbalance in the resisting (hydrostatic confinement) versus driving (positive pore water pressures) forces thereby drives a frequent oscillation of bank stability between stable (at high tide) and unstable states (at low tide). This transition between stability and instability is found not only on a semidiurnal basis,

but also on a longer timeframe. In the spring to neaps transitional period, banks experience the coexistence of high degrees of saturation due to the high spring tides and decreasing confinement pressures favoured by the still moderately high channel water levels. This transitional period creates conditions when failures are more likely to occur.

## 1 Introduction

Rising sea level and increased storminess, driven by climate change, pose significant risks to coastal communities due to their

increased exposure to flooding and erosive events. For example, it has been estimated that 17% of coastlines in the British Isles, and almost 20% in Ireland, are being affected by the combination of sea level rise and the increased frequency of severe storms (MCCIP, 2020). In total, the UK coastline is 17,381 km long, 17.3% of which is experiencing erosive trends (EUROSION, 2004) with the yearly costs of coastal erosion in the UK rising to a possible £126 million per year by 2080 (Foresight, 2004).



The evolution fine-grained shorelines within estuaries are closely connected to bank retreat processes (Zhang et al., 2004, 2021; Guo et al., 2021; Zhao et al., 2022). The growing risks of shoreline erosion have been a cause of concern in recent years. While beach retreat (Jolivet et al., 2019; Bain et al., 2016; Hird et al., 2021; Carvalho and Woodroffe, 2021; Masselink et al., 2016) and cliff erosion (Brooks et al., 2012; Leyland and Darby, 2008; del Río and Gracia, 2009; Young et al., 2014; Hackney et al., 2013) have been well researched, sensitive estuarine environments have received less attention despite the importance

of such settings for human beings. About 60 % of the world's population is concentrated along coasts, and 22 of the largest cities on Earth are located beside estuaries (Harris et al., 2016). Well-preserved estuarine sub-environments such as salt marshes, are essential in the mitigation of coastal flooding, attenuating the wave activity (Möller et al., 2014; Fairchild et al., 2021; Leonardi and Fagherazzi, 2015), but also play a fundamental role in the processes of carbon sequestration and storage (Li et al., 2022; Pendleton et al., 2012), mitigating the effects of global warming. Some studies (Bendoni et al., 2014; Mel et

al., 2022; Carniello et al., 2009; Marani et al., 2011; D'Alpaos et al., 2007) have explored marsh retreat behaviours in microtidal settings (e.g. Venice Lagoon, Italy), while others (Shimozono et al., 2019; Roy et al., 2021) have investigated the factors controlling erosion in large tidal-dominated estuaries, typically emphasising hydrodynamic processes or focusing on soil creep in mesotidal environments based on seasonality-related modifications (Mariotti et al., 2019). In contrast, studies that consider the problem of bank collapse geomechanically, and with a particular focus on hypertidal environments, are lacking. Given the

centrality of estuaries as transitional zones between sea and land, a more complete understanding of the sources, mechanics and rates of bank erosion due to geomechanical failure is of substantial importance. Bank failure processes leading to severe erosive trends in tidal settings have to date been poorly studied and quantified (Gong et al., 2018; Zhao et al., 2022, 2019), especially when compared with the large literature on bank erosion in non-tidal (fluvial) environments (e.g. Rinaldi and Nardi, 2013; Nardi et al., 2012; Patsinghasanee et al., 2018; Julian and Torres, 2006; Darby and Thorne, 1996b; Darby et al., 2000;

Majumdar and Mandal, 2022; Zhang et al., 2021; Thorne and Abt, 1993; Darby et al., 2010; Duong Thi and do Minh, 2019). Given the additional complexity of the process mechanics involved in tidal settings, arising mainly from the presence of bi-directional flows, process insights gained from studies of fluvial bank erosion may not necessarily be transferable to estuarine contexts. The present study seeks to address this gap through an investigation in which a combination of field observations and geotechnical modelling is employed to elucidate the bank failure processes operating in a hypertidal environment (the

Severn estuary, UK).

## 2 Methods

### 2.1 Study site

Hypertidal environments are defined as environments where tidal ranges exceed 6 m (Archer, 2013; Wolanski and Elliott, 2016). Such large tidal ranges usually are attributable to the specific local planform geometry and bathymetric profile (e.g.,

Turnagain Arm in Alaska, Severn Estuary in UK, Cobequid Bay and Salmon River Estuary in the Bay of Fundy, Canada) that favours tidal amplification (Pye and Blott, 2014). The Severn Estuary in south-west Britain (Fig. 1a) is a long funnel-shape

Earth **Surface**
**Dynamics**
Discussions
EGU
estuary characterised by one of the largest tidal ranges, with mean spring tides exceeding 12 m at Avonmouth (port of Bristol), which is located in the outer part of the system.

There are various designations that classify the Severn Estuary into distinct zones. The division adopted here largely is based
on morphological grounds and subdivides the system into outer, middle, and inner estuary segments (Allen, 1990). While the outer estuary is exposed to strong wind-waves forming in the Atlantic and has a substantial width with large silty-sandy bars detached from the banks, the middle estuary is narrower resulting in a more sheltered environment typified by a more uniform width with elongated, semi-detached, bars. The inner part, extending approximately from the largest meander of the system up to the inner tidal limit close to Gloucester, is characterised by a narrow and sinuous single thread channel with bank-attached
bars (Fig. 1b) in which wind-waves are small (< 1 m in height; Allen and Duffy, 1998).

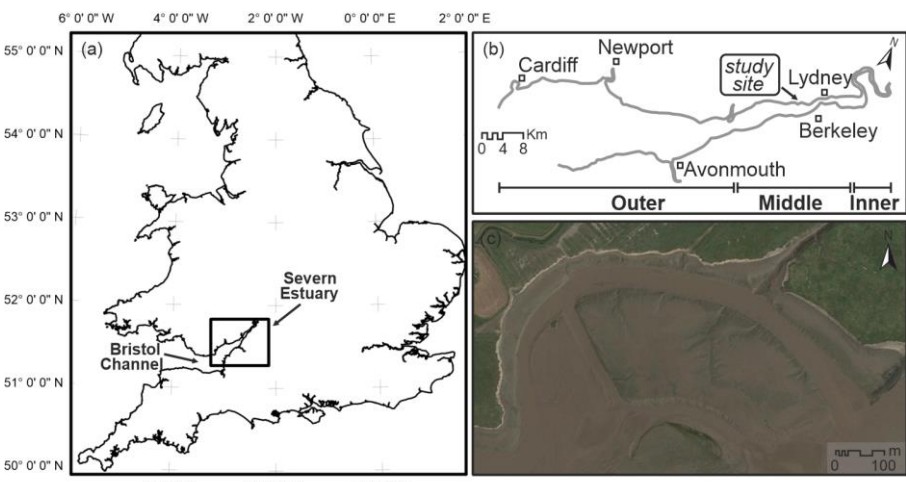

**Figure 1:** Location: (a) The Severn Estuary in south-west Britain (map © Ordnance Survey 2021). (b) Subdivision of the estuary into inner, middle, and outer sectors based on geomorphological features. (c) Google Earth image of the study site in the middle estuary (image © Google Earth July 2021).

Within the Severn Estuary steep banks are widespread, with mass-failure events common. Towards the base of these banks the slopes are typically gentler, and processes (i.e. hydraulic entrainment) other than gravity operate, removing the collapsed material. The Severn estuary receives fine sediments from many sources leading to a stratigraphy that is composed of four discrete lithostratigraphic units (Allen and Rae, 1987) dominated by sequences of silty-clay materials (Allen, 2001). The specific conditions investigated here (Section 2.2) are characteristic of many parts of the Severn Estuary but are not dissimilar
to those that can be encountered in many hypertidal fine-grained estuaries.

**2.2 Mid estuary field surveys**

The focus of this research is centred on a site located in the middle estuary south of Lydney in the Plusterwine area (UK National Grid reference: ST 612995) (Fig. 1b and c) where erosive trends have been well-documented in recent decades (Allen,





2001). To elucidate the detailed mechanics governing these observed mass wasting events, a combination of high-resolution
monitoring and numerical modelling was implemented. A summary of the techniques used for the quantification of the
parameters used in the bank stability modelling, the monitoring periods, and the identification and validation of the bank failure
mechanics contributing to the observed bank retreat are reported in Table 1.

Table 1. Survey types and data collection timeline used in the study site. Major storm events recorded during the monitoring period are
reported [1] Erik, [2] Gareth, [3] Hannah, [4] Atiyah and Brendan, [5] Ciara and Dennis. Note that the Uncrewed Aerial Vehicle (UAV)
survey in February 2018 was used for drone calibration and not for data analysis.
(Met Office Natural events data source, last access on 20 December 2021: https://www.metoffice.gov.uk/weather/warnings-and-advice/uk-
storm-centre/uk-storm-season-2018-19 and https://www.metoffice.gov.uk/weather/warnings-and-advice/uk-storm-centre/uk-storm-season-
2019-20).

| | 2018 | | | | 2019 | | | | | | 2020 | | | |
| --- | --- | --- | --- | --- | --- | --- | --- | --- | --- | --- | --- | --- | --- | --- |
| Surveys | Jan | Feb | Jun | Jul | Feb | Mar | Apr | Jun | Nov | Dec | Jan | Feb | Mar | Sep |
| UAV | | \|--\| | | \|--\| | \|--\| | | | | | | | | | |
| Grainsize | | | \|--\| | | | | | | | | | | | |
| Topographic | \|--\| | | | \|--\| | \|--\| | | | | | | | | | |
| Groundwater | | | | | | | | | \|-----\| | | | \|------\| | | |
| Fix camera | | \|---\| | | | \|----------------\| | | | | | | | | | |
| Geotechnical | | | | | | | | | | | | \|--\| | | |
| Major storms | | | | | -[1]- | -[2]- | -[3]- | | | ---[4]--- | --[5]-- | | | |
| Failures | | \|---\| | | | \|----------------\| | | | | | | | | | |

## 2.2.1 Aerial surveys

The distribution of bank retreat at the study site for the period 2018-2020 was assessed via repeated Structure from Motion
(SfM) surveys undertaken using an Uncrewed Aerial Vehicle (Fig. 2). SfM is a photogrammetric technique that employs
overlapping photographs taken from different viewpoints to obtain a three-dimensional scene (Micheletti et al., 2015) resulting
in a dense point cloud. In the present study, dense point clouds were generated from two aerial surveys (June 2018 and March
2019). These high-resolution (~3 cm at ground level) point clouds were developed from photographs collected with a 3DR
Solo quadcopter equipped with a high-resolution 20.1 MP Sony camera ILCE QX1 and Sigma 28 mm fixed lens. A constant
flight height of 100 m (selected via experimentation across varying flight heights and analysis of the resultant ground pixel
resolution versus image coverage) above ground level was maintained during all the surveys to aid inter-comparison of images
taken at different times during the study. During all surveys the photographs were acquired so that both the overlap and sidelap
between successive images was fixed at 85%, a value found to result in a high degree of precision in matching surface features.
To georeference the images, a variable number (eight on average) of ground control points (GCPs) were placed on the ground
before the photographs were taken by the drone. Each control point target comprised a 1 by 1 m plastic black and white square,





on which a GPS point was recorded using a Leica GS15 base/rover setup with a mean positional accuracy of 0.02 m, allowing precise georeferencing of the entire model during the post-processing stage.

Particular attention was paid to the deployment pattern of these targets by placing a series of GCPs close to the edge of the bank (i.e. the location with the greatest change in elevation in the scene) where potential image artefacts, such as doming (Rosnell and Honkavaara, 2012; Javernick et al., 2014) are most likely. Before proceeding with post-processing, each photograph was inspected manually for quality and any out-of-focus images were excluded from further analysis. The quality of the photographs was analysed using the image quality estimation feature included in the software Agisoft PhotoScan Pro

1.6. Images with a quality value of less than 0.5 units were excluded from further analysis. Bank positional total cumulative errors ($\sum Et$) were evaluated as the square root of the sum of the squares of the different error variables (Kermani et al., 2016; Thieler and Danforth, 1994) such as the ground control markers (GCP) used to georeference the model and the image pixel size:

$$\sum Et^2 \sqrt{E_{GCP}^2 + E_{pix}^2} \qquad\qquad (1)$$

where $E_{GCP}^2$ is the relative accuracy of the ground control point after the photo alignment in the software and $E_{pix}^2$ is the error connected to the pixel resolution (Table 2).

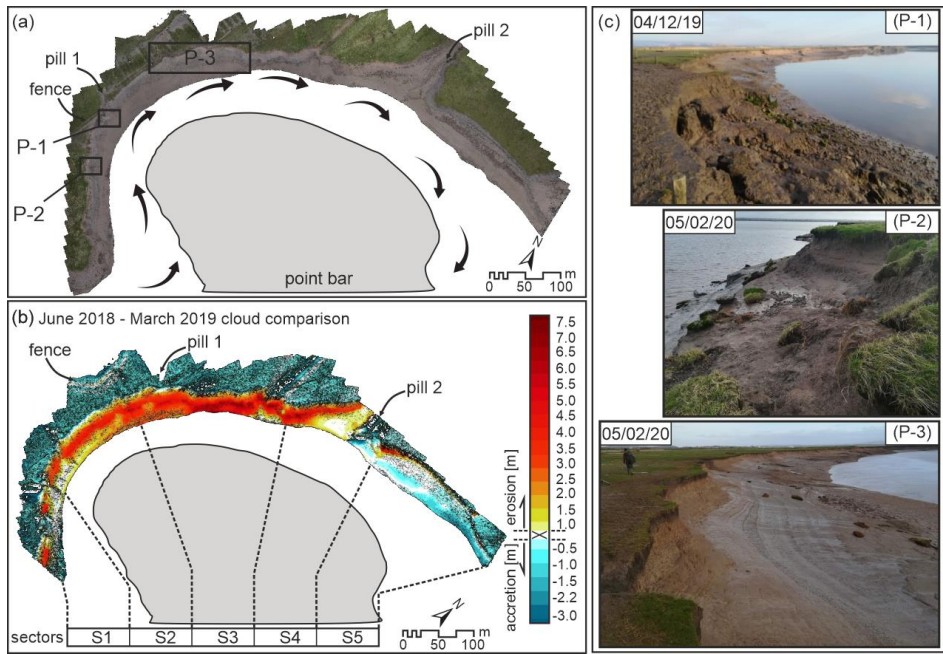

**Figure 2. Embayment details (a) Studied embayment (mosaic of images), with inferred flood tidal flow represented by the black arrows; (b) point cloud difference between the June 2018 and March 2019 surveys and site subdivision into sectors, and (c)**
**photographs of three representative areas of the study site shown in the black boxes on (a).**





The point clouds were integrated into the OSGB 1936 coordinate system and exported into CloudCompare (Lague et al., 2013) for further analyses. In CloudCompare the difference between the clouds was investigated using the Multiple Model to Model Cloud Comparison (M3C2) plugin (Lague et al., 2013). The algorithm applied with this technique measures the distance between the two dense point clouds in the direction normal to the local surface, operating directly on point clouds without

triangulation or gridding (Lague et al., 2013). This method is well-known for maintaining an accurate estimate both in the vertical and horizontal displacement between two scans (James et al., 2017; Barnhart and Crosby, 2013).

Table 2. Estimated total errors for each bank digitisation for orthomosaic images derived from aerial photographs.


| | June 2018 | March 2019 |
|---|---|---|
| Number of photos | 423 | 787 |
| GCP error, $E_{GCP}$ [m] | 0.08 | 0.13 |
| Image pixel size, $E_{pix}$ | 0.175 | 0.175 |
| Total error, $\sum Et$ [m] | 0.19 | 0.21 |


### 2.2.1 Fixed camera monitoring

The second type of monitoring survey consisted of deploying a fixed 12MP camera installed along the mid-western part of the studied embayment during two survey periods (30 January-05 March 2018 and 19 March-28 July 2019) (Fig. 3). The photographs (3,842 images taken at intervals of 1 hour during each of the two acquisition periods), together with the SfM

model and a series of dGPS topographic surveys (Table 1), were used to define the elevation of the bank top (8.4 m), the flood overtopping stage during high water phases, and the bank geometry used in the development of the bank stability model.

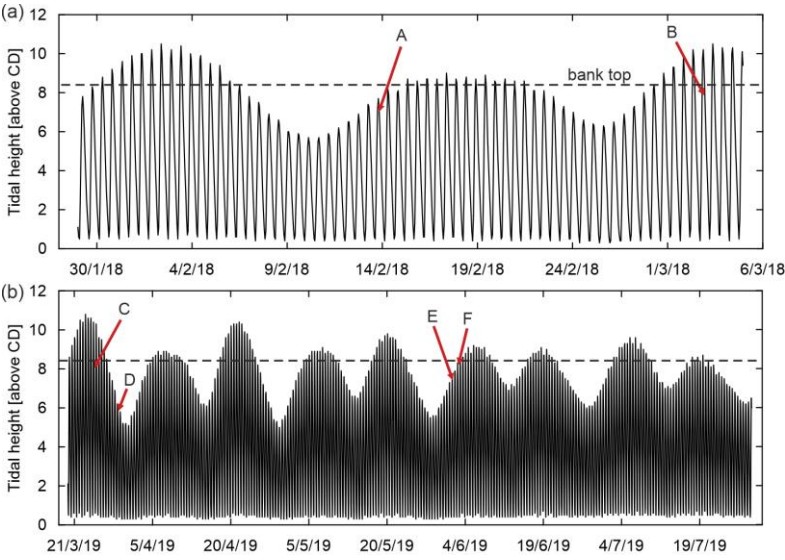

**Figure 3. Tidal cycle for the two fixed camera survey periods: (a) 30/01/2018-05/03/2018; (b) 19/03/2019-28/07/2019 (data from UK Admiralty Total Tide software). Red arrows indicate the most substantial bank collapse events (A to F) identified during the**
**monitoring periods (see Figure 4).**

Earth **Surface**
Dynamics
Discussions

The photographs ware also used to validate the variations in tidal water stages obtained from the Admiralty Total Tide tables and their association with both timing and nature of discrete mass wasting events. A clear identification of all the mass wasting events that occurred during the monitoring period was not possible because minor failures (smaller than ~40cm$^2$) could not be identified on the images. Also, large collapses located farther than 20-30 m from the location of the camera were not

identifiable. In total 25 major failure events were identified and some of the most substantial are shown in Figure 4.

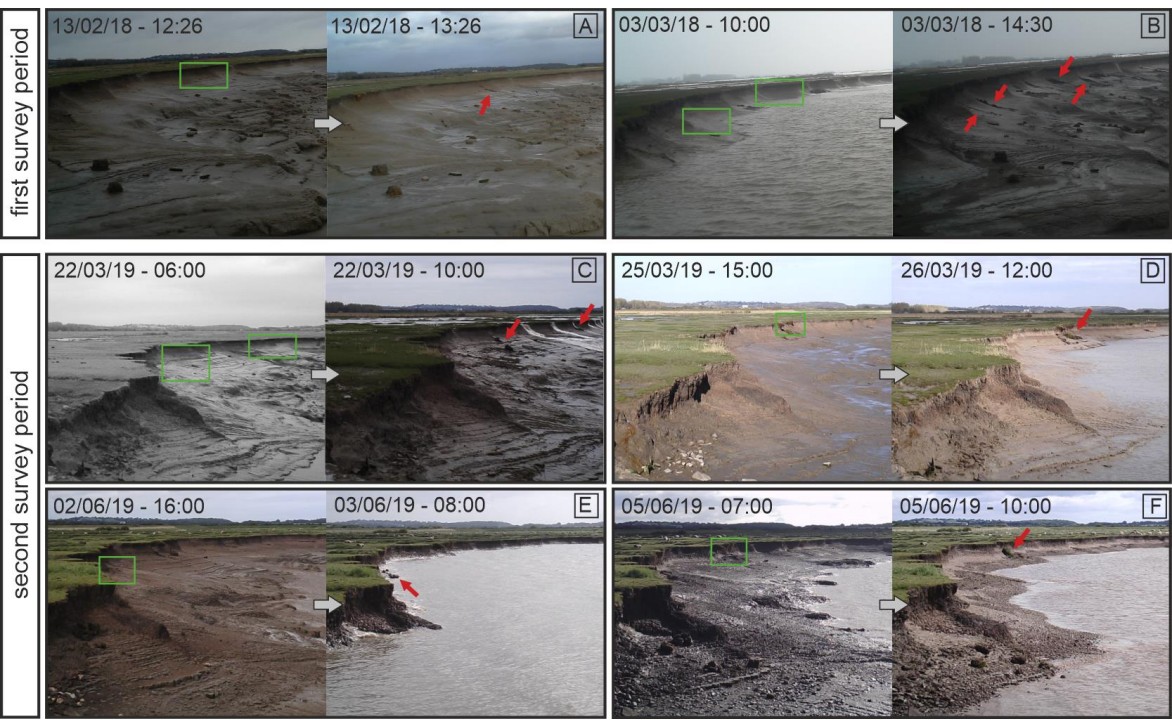

**Figure 4. Time sequences of notable mass failure events identified at the study site during the two periods of deployment of the fixed camera. Red arrows in the photographs indicate the collapse events; green boxes highlight the site of the bank where successive collapses occur. Photographs connected by white arrows are successive (very close) in time. The failure events are also highlighted**

**in the tidal range charts in Figure 3.**

### 2.2.2 Groundwater monitoring system

The impact that tidal variations have on groundwater fluctuations and thus pore-water pressures within the bank materials was assessed by implementing a monitoring system in the central-western part of the study site (Fig. 5) where two monitoring wells (60 mm of internal diameter) were drilled to a depth of 3 m using a percussion auger. Within each well, PVC pipes with a

diameter of 40 mm were inserted and the cavity between the external part of the pipe and the wall of the borehole was filled with fine pebbles to filter the water and prevent the ingress of fine-grained sediment into the well. Water table fluctuations within each well were monitored every 2 minutes using logging pressure transducers (LevelSCOUT probes; absolute depth range: 25 m, pressure in-depth accuracy: ±0.05% FoS (at 20°) during two separate periods (25 November-4 December 2019; and 5 February-3 March 2020) (Fig. 5).





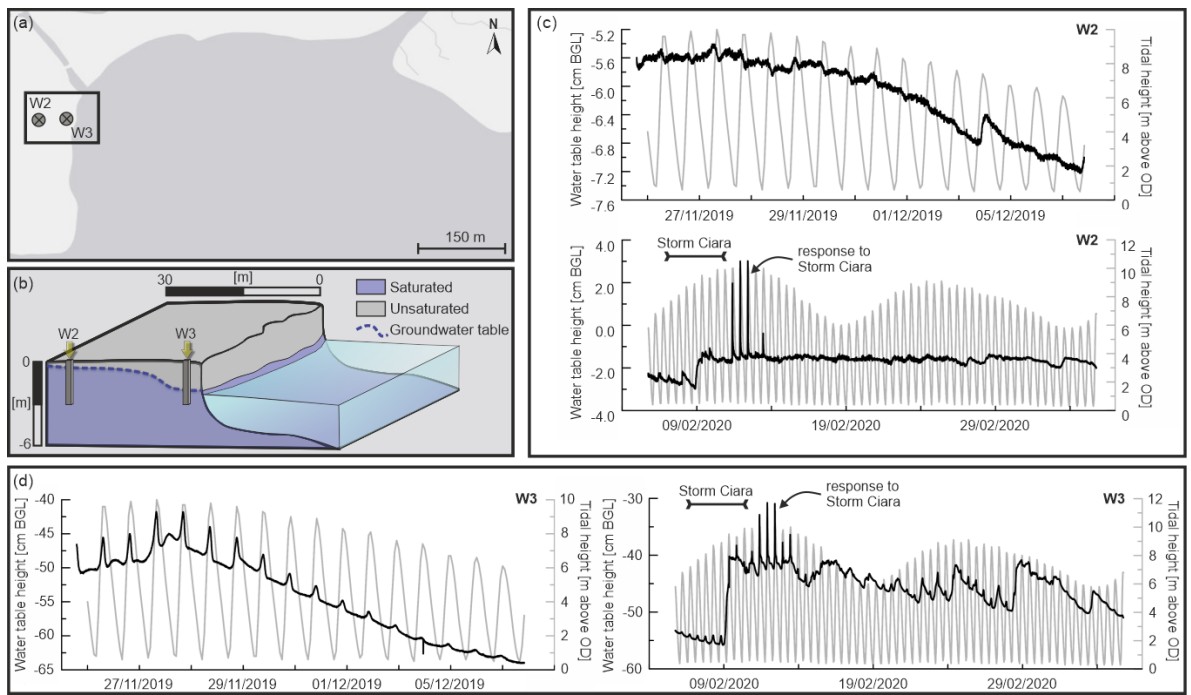

**Figure 5.** Groundwater monitoring: (a) Location of the groundwater monitoring system (image © Esri World Light Gray Base July 2021). (b) Material layering and position of wells on the bank (bank top level = 8.4m above datum). (c) Groundwater table fluctuations for the first and second monitoring periods for the bank distal monitoring well W2. (d) and the bank proximal monitoring well W3. High peaks in the water table level present in the February/March 2020 plots are linked to storm Ciara which effected western UK in early February 2020. The two monitoring wells (W2 and W3) were used in the modelling. (Storm data retrieved from https://www.metoffice.gov.uk/weather/warnings-and-advice/uk-stormcentre/ uk-storm-season-2019-20).

### 2.3 Model setup and design

Identification of the mechanisms responsible for driving episodes of bank failure observed during the monitoring periods was obtained through a detailed bank stability analysis. To develop the stability analysis, the geotechnical properties of bank material samples taken from the study site were evaluated in the laboratory via consolidated-undrained triaxial shear testing (British Standard Institution, 1990), and the model was setup and validated using the groundwater data together with topographic surveys. The results presented here provide insight into the detailed processes leading to the observed mass wasting events.

Preliminary tests, that aimed to characterise the density of the soil samples under dry and ambient conditions, were conducted on multiple blocks of sediment extracted from two layers (Fig. 6) of the bank at three locations in the central part of the study site. The two layers from which samples were extracted were defined based on observations of the stratigraphic sequences, combined with detailed observations of changes in grain size properties as measured in the cores extracted from the monitoring well W3 and from the bank face. The two maximum effective stresses $\sigma'1$ and $\sigma'3$ were calculated from the triaxial tests and





used for the bank stability model, the latter being performed using the GeoSlope software suite (GeoSlope International Ltd, 2018). The hydraulic conductivity ($K$) for each layer was obtained via application of the Kozeny-Carmen equation (Freeze and Cherry, 1979; Rosas et al., 2014), to infer the conductivity based on grain size:

$$K = C_{KC} \frac{g}{v} \frac{n^3}{(1-n)^2} D_{10}^2 \qquad (2)$$

where $K$ is the hydraulic conductivity [m s$^{-1}$], $C_{KC}$ is an empirically based coefficient taken here to equal to 1/180

[dimensionless], $g$ is the gravitational acceleration [9.8 m s$^{-2}$], $v$ is the kinematic viscosity of water (in this case, a value of 1.1500e$^{-6}$ m$^2$ s$^{-1}$ was used to represent brackish sea water), and $n$ is the total porosity obtained from the laboratory experiments [dimensionless], and $D_{10}$ is the 10% cumulative passing (geotechnical grain size distribution) [mm]. The data resulting from the geotechnical parametrisation and subsequently used in the model are reported in Table 3.

Table 3. Geotechnical properties of the bank materials at the study site. $\theta s$ and $\theta r$ [m$^3$ m$^{-3}$] indicate the saturated and residual water content

respectively, $c$ cohesion [kPa], $\Phi$ = friction angle [°], $Uw$ = unit weight [kN m$^{-3}$], $K$ = hydraulic conductivity (average) [m s$^{-1}$], and $Dx$ = geotechnical grain size distribution [µm].

|  | $Uw$ | $K$ | $\theta r$ | $\theta s$ | $\Phi$ | $c$ | $D_{10}$ | $D_{50}$ | $D_{90}$ |
|---|---|---|---|---|---|---|---|---|---|
| Upper bank | 30 | 5.88 x10$^{-8}$ | 0.07 | 0.35 | 7.2 | 3 | 2.03 | 10.5 | 37.0 |
| Lower bank | 52 | 3.14 x10$^{-8}$ | 0.07 | 0.45 | 2.3 | 7 | 1.93 | 11.8 | 40.0 |

The bank stability simulations were developed by coupling a 2D finite element seepage analysis (to simulate the evolution of

the pore water pressures in the simulated bank) and a limit equilibrium stability analysis based on the Morgenstern-Price method (Morgenstern and Price, 1965). The model setup procedure comprised four steps: (i) the delineation of a simplified geometry to represent the bank morphology (based on the topographic surveys and SfM model of the study site); (ii) the discretisation of the different sedimentary horizons of the bank into different layers based on their geotechnical properties; (iii) the establishment of the time-varying boundary conditions that force the model simulations, here represented by the water

level fluctuations recorded at Berkeley (Fig. 1b); and (iv) the temporal integration of the transient analysis. Following this process, the investigated bank was discretised into a c. 0.5 m resolution irregular triangular grid composed of 1792 elements subdivided into two bank material layers (Upper and Lower bank deposits) (Fig. 6c). The different parameters used to represent the properties of the materials in these two layers are reported in Figure 6a.

It is important to note that the simulations conducted here are affected only by tidal conditions. Given the fact that the impact

of rainfall is very low if compared to the very large tidal fluctuations, the influence of pore pressure variations driven by infiltrating rainfall is neglected in this study.



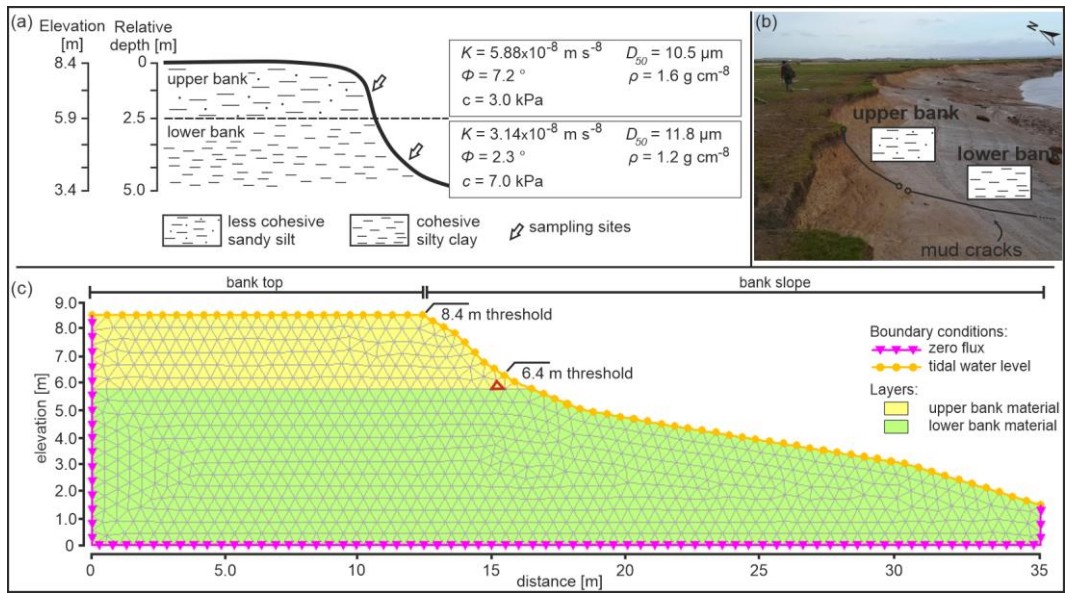

**Figure 6. (a) Representation of the investigated bank at the study site illustrating the most important geotechnical characteristics of the two sedimentary layers ($K$ = conductivity, $\Phi$ = angle of internal friction, $c$ = cohesion, $D_{50}$ = geotechnical grain size distribution, and $\rho$ = dry bulk density). (b) Photograph of the central part of the study site showing the progressive change in grain size. (c) Representation of the mesh characteristics used in the bank stability model. The cell used to represent the temporal variations of the pore water pressure (see Figure 9 and 10) is highlighted in red. The different boundary conditions employed in the simulations are indicated. Note that the boundary condition indicated in yellow indicates the surface at which the tidal level interacts with the bank (with the entire bank slope in this specific example of high tide).**

The model was driven with time-varying boundary conditions along the fringes of the finite element grid. For the gently degrading bank slope and the vertical face of the bank, a total head versus time function was used to represent the tidally driven oscillations in water level. Given the repetitive nature of the overbank inundation of the study site, the same total head versus time boundary conditions were also applied to the nodes along the bank-top profile. A null-flux condition was applied along the remaining vertical and horizontal boundaries.

Note that a series of sensitivity analyses were carried out to ensure the robustness of the model setup process. Specifically, these sensitivity tests were designed to demonstrate that the simulations are independent of variations in the discretisation of the selected finite element grid, as well as of variations in those model boundary conditions that were specified based on estimated values rather than measurements. Regarding the latter, these estimated boundary conditions refer specifically to the zero flux conditions assigned to the left lateral and basal horizontal boundaries. The sensitivity tests revealed that the simulated 255     pore water pressures within the materials close to the bank face, which are subject to the investigated bank collapses, are independent of the assumed zero flux conditions. Similarly, comparisons between a coarser and a more refined mesh indicate that the model results are insensitive to the discretised grid design selected here.

Two separate simulation scenarios were defined (Fig. 7). The first scenario employs a 10-minute time increment for a shorter duration simulation (22 March 2019 06:00 – 23 March 2019 08:00), and a second scenario simulates the entire calendar year


Earth **Surface**
Dynamics
Discussions

of 2020 at a 1 hour time-step. The first scenario was selected to ensure an overlap with the field monitoring observations and

focused on seeking to understand the factors (these factors include gullying, basal removal, and tidally-driven fluctuations in

pore-water and hydrostatic confinement pressures, as summarised on Fig. 8) triggering the observed mass wasting events in

this specific period and as a form of model validation. The initial groundwater table conditions were defined using the water

table level from the bank-edge proximal monitoring well W3 data from November 2019, when conditions were not dissimilar

to those in March 2019. The second simulation scenario was used to study the factors influencing seasonal variations in bank

stability (as indicated by the Factor of Safety, here after FoS). This second simulation scenario was initialised based on water

table conditions observed in the monitoring well W3 in February 2020 and was forced using the time-varying tidal water levels

as observed at Berkeley.

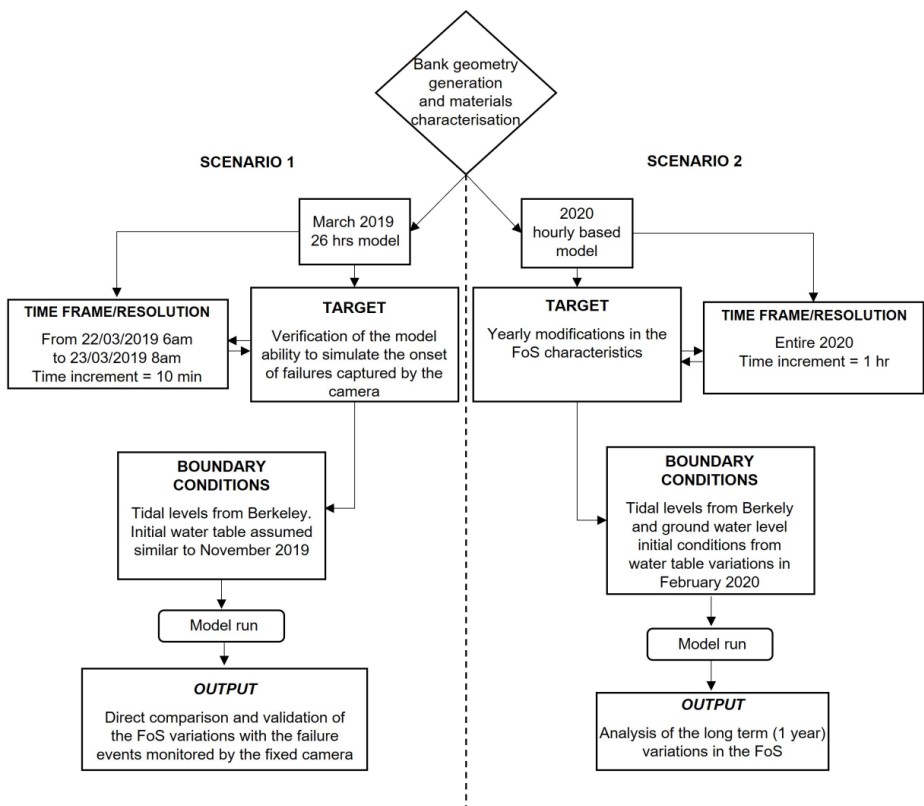

**Figure 7. Summary of the two modelling scenarios.**

In interpreting the results of the model simulations it may be noted that, based on topographic and SfM surveys, two important

topographic controls influence the behaviour of seepage flows into and out of the bank, as well as the hydrostatic confining

pressure exerted by water against the bank face (Figure 6). Specifically, the elevation at which the flow in the channel overtops

the bank (allowing potential seepage into the bank from the floodplain surface) is 8.4 m at the study site, whereas the 6.4 m

elevation represents the limit below which the tidal waters no longer exert any direct influence on the simulated incipient





failure bock (vertical face of the bank). Also, note that the bank geometry remains constant throughout all the simulations reported here, even after mass failure events are predicted (FoS<1). This constrained geometry is a recognised limitation because in reality, after the incipient failure block is translated downslope, the post-failure bank surface (usually of the order of 35-45°) will be much less steep than prior to failure (typically ~85°). Thus the model is conservative as the factor of safety

after simulated bank collapse events would, in reality, be higher (i.e. more stable) than simulated here.

## 3 Results

### 3.1 Field surveys

Point cloud comparison analysis (Fig. 2) highlighted intense bank mobility across the whole study area, both in terms of erosion and accretion. Areas dominated by accretion are mainly localised within sectors S4 and S5 (Fig. 2b), which are located in the

eastern portion of the study area, a zone that is most protected from the tidal currents. This protection is granted by the presence of coastal defences (rock armour made by large boulders) at the eastern limit of the embayment, and by the presence of a large sand bar in the centre of the embayment that acts to protect and deflect the flows (Fig. 2a) decreasing the erosion and, in turn, favouring the accretion and the redistribution of the sediments delivered by the gully (gully 2 in Fig. 2a) sited in the central-eastern part of the site. The deflection of the tidal-flood currents may also help to explain the intense erosion identified in

sectors 1-4 (Fig. 2b, central and western part of the site), where bank retreat of up to 7 m a$^{-1}$ was identified. Here, erosive events mainly are caused by mass wasting. Hydraulic entrainment is a factor contributing to the removal of deposited sediments from the toe of the bank slope. In the central part of the study site, the erosion is spread along the entire bank (this is the area with the highest levels of bank retreat), while on the western portion of the site erosive trends mainly are localised in the top part of the bank. This spatial pattern may reflect the distribution of flow in the study area and also is reflected in the presence

of several collapsed blocks (Fig. 2c) along the bank slope (not present in the central part of the embayment where slumped blocks are quickly removed), which may act to provide partial protection against further erosion. Slump blocks have been confirmed as having the effect of reducing erosion rates in other bankline environments (Motta et al., 2014; Hackney et al., 2015; Thorne and Tovey, 1981; Wood et al., 2001).

Through the deployment of a fixed camera in the central part of the study site, 25 failure episodes in the bank zone close to

(distance <30 m) the camera location were identified, enabling analysis of the type of mechanisms involved (Fig. 3 and 4). It is evident that the identified bank failures all take place during the receding phase of the tidal cycle. Specifically, seven collapses occurred on the receding stage just after the high water peak, while the other 18 occurred later, closer to the low water stages. The largest collapses are all linked to very high tidal stages overtopping the bankline (i.e. tidal peaks >8.4 m). During the tidal ebb phase following these high tides, the bank is susceptible to the combined effects of two forces that promote

failure: (i) the combined force of internal pore water pressure of the saturated bank materials and the lack of a holding pressure, and (ii) the presence of a hydraulic flow that is activated during the initial stages of the tidal recession phase when the tidal



waters drop below the bank edge, triggering gullying along the floodplain margin. These intense drainage flows are clearly visible on some of the images (Fig. 4 photograph C) and they not only prepare the face for successive large block failures, but they also directly affect the detachment and transport of bank materials toward the bank toe (Fig. 8). Gullies, such as those

observed here, have been mentioned before in earlier studies of the Severn Estuary (Crowther et al., 2008) but they were never directly recorded and linked to other erosive mechanisms in the bank destabilisation process. The photographs reveal that such gullies overlap with other failure mechanisms such as toppling and cantilever bank failures (Fig. 8); toppling occurs in the earlier stages of the collapse process when the top-portion of the bankline slides along the critical slip surface during the tidal ebb phase; in a second phase of the collapse, the bankline is characterised by a reshape of its face whereby the uppermost part

of the bank slope forms a cantilever. This specific style of failure likely reflects the specific stratigraphy of the studied area where two layers of fine material are stacked. The nature of the layers, and specifically the positioning of the incipient failure plane at the interface between the layers, suggests that these stratigraphic discontinuities can play a key role in the failure modalities.

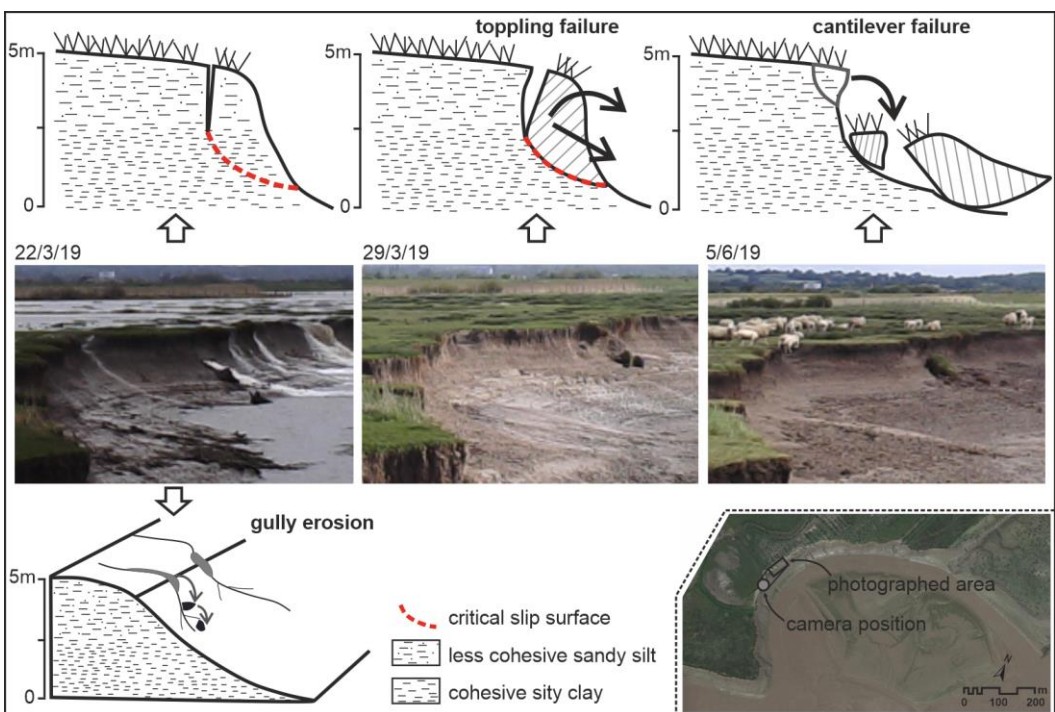

**Figure 8. The typical collapse mechanisms observed at the study site (note that the photographs indicated refer to those presented in Figure 4). Image © Google Earth July 2021.**

The ground water fluctuations recorded during the monitoring periods (Fig. 5) confirm that the sediments of the studied bank have a relatively low level of transmissivity such that the groundwater response to the tidal oscillations is damped relative to the variations in hydrostatic confining pressure driven by the tidal level fluctuations. While the distal monitoring well (i.e. W2)

is much less affected by variations in the tidal water level attaining an average water table elevation (below local ground-level)



of c. 6 cm, the bank-proximal monitoring well (i.e. W3) shows a higher responsiveness to tidal oscillations (average water depth 50 cm). The system is therefore affected by tidal variations predominantly in the bank face fringing zone (within a distance of 3-5 m from the bank face).

## 3.2 Bank stability model

The Scenario 1 simulation results (Fig. 9) represent a very high-resolution model (10 minutes time increment) that provides a detailed visualisation of the bank failure processes observed in March 2019 during a 1-day monitoring window. In contrast, Scenario 2 focuses predominantly on the semidiurnal nature of the tidal cycle in the mid Severn Estuary, studying variations in stability over a much longer timeframe (the calendar year 2020), which allows for the effects of variations associated with the differences between spring and neap tidal cycles to be taken into consideration. These results were compared with and

combined with real-world observations (from the fixed camera photographs) leading to a detailed analysis of the failure mechanics. The evolution of the bank was subdivided into phases (denoted P1, P2, P3, etc), and some additional specific time points corresponding to specific exemplar photographs were marked. Figure 9 (panel a) sketches the tidal cycle represented in the model; the different boxes exhibit different colours highlighting the relative Factor of Safety (FoS) of the bank. In the last panel of Figure 9, some images linked to specific moments in the simulation have been further analysed with three

supplementary zooms related to the pre-failure ($t_0$), failure ($t_3$), and a secondary failure phase (additional minor collapses) ($t_4$).

In the first part of phase P1 (Figure 9b point A, corresponding to photograph $t_2$ on the last panel), the tide is at peak level and the entire tidal flat is inundated (22 March 2019, 07:30-08:00). The groundwater table is therefore almost at ground level. Under these conditions, even though the elevated positive pore pressure (~43 kPa at 08:10) promotes bank instability, the simulation results indicate a rising FoS (i.e. increasing stability) (from a value of 0.77 at $t_0$ to a value of 1.14 at 07:20) due to

the high hydrostatic confining pressure (~100 kPa). After the peak in the tidal level is surpassed (mid-phase P1 at 08:40), the pore water pressure also begins to decline. Thus, even as the hydrostatic confining pressure is reducing with the retreat of the tidal waters (106 kPa at the peak at 08:30, 100 kPa at 08:50), the soil saturation remains high (~40 kPa at 08:50) favouring the decrease in the FoS (FoS = 1.07). Toward the end of phase P1 (c. 09:40), the retreat of the tidal waters generates a considerable and fast reduction in the hydrostatic confining pressure (from ~100 kPa to ~88 kPa at the end of P1). This rapidity, together

with the elevated pore water pressure associated with the earlier rising tide that led to a saturation of the bank deposits (the peak in the saturation corresponds to the peak in tidal level), creates a condition in which the bank begins to experience a decline in the FoS (albeit the bank is not yet unstable).

The decrease in the FoS continues also in phase P2. From point B onwards, the stabilising effect of the hydrostatic confining pressure exerted by the flow in the tidal channel is rapidly reduced as the tidal level starts to fall below the elevation of the

bank top (i.e. once it falls below the 8.4 m elevation of the bank top) and the FoS drops rapidly, changing the bank from a stable to unstable state (from a FoS of 1.05 at 09:45 to 0.9 at c. 10:00; photograph $t_3$). During this phase the pore water pressure





remains positive (falling from 33 kPa at 11:40 to 13 kPa at 11:55) but the condition of instability is further exacerbated by the elevation of the ebbing tide falling below the 6.4 m incipient failure toe elevation threshold (point B1), causing the hydrostatic confining pressure to fall to a value of zero. Therefore, conditions during this phase (P3) are key for promoting bank instability

because destabilising forces remain high and diminish only gradually whereas the stabilising forces are removed. In Phase P3 (from 11:50 to 19:20), the elevation of the water in the channel remains below the 6.4 m threshold that maintains the stabilising confining pressure at zero until the elevation of the water surface exceeds 6.4 m on the next rising tide (starting at point D at 19:20). With zero confining pressure, phase P3 has the lowest simulated FoS of the entire simulation. During this phase, the bank internal pore pressure evolves only gradually as slow seepage outflow (typical velocities $3.5 \times 10^{-10}$ m s$^{-1}$) gradually drains

the bank, declining from a value of ~13 kPa to ~10 kPa between 11:50 and 19:20, respectively. The bank FoS is, therefore, unable to drop significantly further during this period because the new rising tide terminates phase P3 and induces a new stage of rising bank stability.

Between points D (19:20) and E (21:00), with a swiftly rising tide, the pore pressure again increases due to the inwards-directed seepage flow (typical seepage velocities of $3 \times 10^{-8}$ m s$^{-1}$) but the hydrostatic confining pressure rises more rapidly from ~62

kPa at the start of P4 to ~100 kPa at 21:00 giving a rate of change of ~4 kPa hr$^{-1}$ (compared with the pore pressure curve gradient c. 1 kPa hr$^{-1}$). The very low FoS value persists until the tide again approaches the 6.4 m threshold (point D in Fig. 9b). After point D (19:20), the increase in hydrostatic confining pressure dominates over the slowly increasing positive pore water pressures induced by the new phase of seepage inflow, causing the bank stability to increase to a FoS value of 1.1 at the start of phase P4 (19:35). A second collapse event is predicted in the last phase (P5, which lasts from 23:20 to the end of the

simulation), which corresponds to the next ebb tide. In this second ebb tide phase, the simulated FoS drops below 1 a few minutes after the water level falls below the bank top threshold (22:50); the confinement pressure is no longer present along the top portion of the bank while the pore pressure is decreasing following the recession of the tide but is still high (~34 kPa). Similar to phase P2, the most unstable condition is reached when the hydrostatic confining pressure is lost but the positive pore water pressure within the soil remains relatively high (the hydrostatic confining pressure starts dropping from ~62 kPa at

23:40 to zero at 23:50 when the pore pressure is still ~8 kPa and the FoS is 0.75). Note that phases 4 and 5 both occurred during hours of darkness and the fixed camera was therefore unable to yield images that could constrain precisely the observed timing of the predicted failure. Nevertheless, it is noteworthy that a new mass wasting event was apparent in the first image captured in daylight after the simulated collapse (photograph $t_4$, taken at 08:09 on 23 March 2019, Fig. 9d).

It should be noted that the model simulations are forced using water-level data from Berkeley, which is located ~5 km up-

estuary from the study site. This locational discrepancy means that: (i) the actual tidal wave would be experienced ~30 minutes earlier at the Plusterwine field site than in the model and (ii) there will be a slight mismatch (underestimate) in the tidal levels used to force the model versus those actually experienced at the study site. It may, therefore, be expected that the model simulated FoS is both slightly underestimated (i.e. simulated FoS values are lower than in reality) and out of phase (the




Earth **Surface**
**Dynamics**
Discussions

simulated FoS would precede the actual FoS trend by ~30 minutes) with the true factor of safety. Nevertheless, it is instructive

to note that the observed bank failure occurs in the four-hour window between images $t_2$ and $t_3$ on Figure 9 (panel b) during

the period in which the simulated FoS declines below the critical value of 1, lending confidence that the model is correctly

predicting the failure mechanics. Overall, the model reproduces the overall dynamics of the bank stability response quite well,

albeit with the slight offset in the value and timing of the FoS curve.

**Figure 9. Simulated and observed (from fixed camera images) bank stability conditions at the Plusterwine study area during a 26**
**hrs time lapse (22-23 March 2019). (a) Cartoon of the studied bank and related conditions; (b) physical conditions of the bank. The**
**letters refer to stages of evolution indicated on panel (a) and markers to photographs in panel (c).**



While the results of the first scenario focused predominantly on the semidiurnal nature of the tidal cycle in the mid Severn Estuary, the results of the second scenario (Fig. 10) focus on fluctuations in stability over a longer timeframe, which allows

for the effects of variations associated with the differences between spring and neap tidal cycles to be taken into consideration. This simulation (Fig. 10) represents the whole of the year 2020 at a temporal resolution of 1 hour, but with a detailed focus on the period of February-March 2020, as this is the time interval that matches the groundwater table monitoring window. Through this entire 1-year modelled timeframe, it is possible to appreciate how the FoS oscillates above and below stable conditions (FoS>1, FoS<1, respectively). Even though the fine-grained, cohesive, nature of the alluvial deposits should promote a high

degree of stability, the simulated bank is clearly close to the critical condition. This marginal stability status appears broadly consistent with the observations made with the field surveys that highlight the erosive activity in the study area.

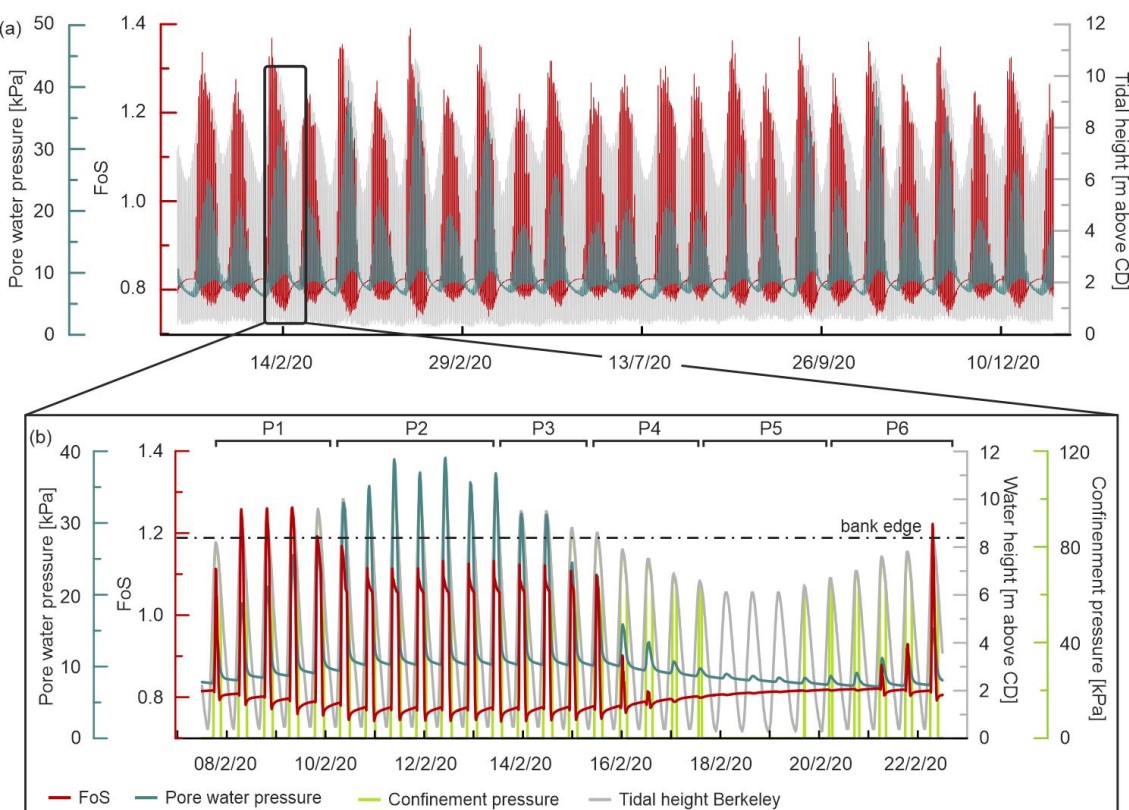

**Figure 10. (a) Computed bank stability analysis for the year 2020 (b) Zoom into the period covered by the groundwater table monitoring in February 2020. This second plot is subdivided into different Phases (P1, P2, P3, etc) and pore water pressures are**
**plotted for a representative cell the location of which is highlighted on Figure 6.**

To better describe the long-term effects of the destabilising forces and their mutual interactions with the stabilising forces, the dynamics occurring over a period of 16 days (07 February 2020 12:00 – 22 February 2020 12:00) is represented (Fig. 10b) and discussed. In this longer duration analysis, Phase P1 covers a time period in which the tidal conditions are moving from neap to spring tide, therefore from a state where the bank top is not subject to daily inundations (the high waters only lap the





bank edge) toward a state where the bank is regularly flooded (peak tides higher than 8.4 m). In P1, the FoS fluctuates in response to the variations in the hydrostatic confining pressure (driven by the tidal water body in the channel) but with a progressive rise as the tidal stage moves toward the spring phase (reached in P2). At the onset of this phase P1, the FoS is still critical (~0.82 – low water stage) as a consequence of the previous drawdown stage but is rising due to the general increase in water level. In the mid-period of this phase, the bank eventually becomes stable. In such circumstances (core of phase P1), the

hydrostatic confining pressures exerted by the tidal waters (~82 kPa) are relatively high and dominate over the internal pore water pressure of ~18 kPa. During P1, the bankline is subject to a regular alternation of stable conditions during high water stages and unstable phases during low water stages when the FoS drops below 0.8. As with the results of simulation Scenario 1, this pattern of simulated bank stability is explained by the increasing hydrostatic confining pressure exerted by the tidal water tending to dominate over the slow (destabilising) and damped variations of the pore water pressure.

In phase P2, the FoS remains above a value of 1 during high water stages but experiences a gradual decline. This behaviour occurs because of the superimposition of the pore water pressure oscillations at the scale of a (non-semidiurnal) spring-neap tidal cycle onto the semidiurnal oscillations. This superimposition leads to a condition in which the destabilising pore water pressure attains higher values than in the previous phases (~27 kPa at the end of phase P1; ~40 kPa at the mid-point of P2). In such circumstances, the slow response of the pore water pressure (due to the low hydraulic transmissivity of the alluvium),

causes the high pore pressures to overlap with the apical part of the spring tide conditions (core of phase P2), such that the stabilising effects of the hydrostatic confining pressure no longer dominate the now very high (~40 kPa) pore water pressure levels, diminishing the stability of the bank (with FoS values approaching 1.1 during high water stages).

Moving toward neap tide conditions (end phase P2 and during phase P3) the bank is no longer experiencing bankfull stages and therefore no more full saturation conditions are present. Neap tides water levels decrease the bank saturation but are still

sufficient high to stabilise the bank materials (confinement pressure) leading to a condition where the confinement pressure is more efficient than the pore pressure destabilisation. Therefore, the bank is still experiencing a degree of stability.

In phase P4, the bank is influenced by the neap stage with an important reduction in the tidal heights. The poorly conductive sediments are still experiencing high pore water pressures (~10 kPa in the central part of P4) even if the tidal level is lower. In this phase, the hydrostatic confining pressure regularly falls to zero during low water stages (largely reducing the stabilisation

effect of this component), favouring factor of safety values below 1. At the onset of phase P5, the bank is still experiencing a condition of instability because of the slow process of dissipation of the pore water pressure (residual positive pore pressure ~8 kPa) and the absence of hydrostatic confining pressure (tidal waters still regularly remain below the 6.4 m threshold). Notwithstanding this, in P5 the overall trend is one of a slowly rising FoS before the onset of a new spring tide condition, because of the gradual dissipation of the pore pressures (~6 kPa in the last part of P5). During the initial part of P6, the tidal

waters shift back toward a spring condition causing a rise in the hydrostatic confining pressure while the pore water pressure





curve is still declining. Once the new spring tide condition is fully set (at the end of phase P6), the pore water pressure again is dominated by the hydrostatic confining pressure, so that the bank returns to more stable conditions (FoS>1.2).

## 4 Discussion

The results illustrate how it is the interplay between the destabilising pore water pressure and the stabilising hydrostatic

confining pressure that determines variations in the stability of a bank in tide-dominated settings. This interaction already has been deemed as being important in riverine environments (Rinaldi et al., 2004; Dapporto et al., 2001; Darby and Thorne, 1996a; Springer et al., 1985; Twidale, 1964; Lawler et al., 1997). The roles of alternative weakening factors (such as rainfall, frost and, dehydration), is considered to be much lower than the effect of the very large tidal fluctuations present in the study area and therefore are excluded from the present model. The model presented here indicates that the semidiurnal oscillations

between high and low water create conditions under which steep banks become unstable during drawdown stages when the hydrostatic confining pressures are removed and the bank materials are still completely saturated. These conditions are compounded especially during spring tides that overflow the bank top and which thereby initially stabilise the bank (FoS constantly >1) due to the high confining pressures, but subsequently favour the development of high pore water pressures during ebb tides. Moreover, the overbank recession phase also favours gullying through drainage of water from the bank top,

leading to additional instability, particularly in the top portions of the banks. During neap tidal periods, on the other hand, the bank is subject to conditions in which the tidal waters do not inundate the floodplain and the destabilisation of the bank is driven exclusively by the interplay between the (lower) positive pore pressures and the lack of hydrostatic confinement during ebb tides. A key component of the presented analyses is the presence of a temporal lag between the pore water pressure level and the tide height (confinement pressure) before the onset of failures. These differences mean that during the ebb phases that

follow high tides, the very high degree of saturation in the bank materials combined with their low hydraulic transmissivity maintains relatively high and gradually declining pore water pressures, whereas the hydrostatic confining pressure reduces much more rapidly during the falling tidal level. This control results in the most unstable conditions occurring in the transition period from high to low water stage. This transition between stability and instability is found not only on a semidiurnal basis, but also on a longer timeframe. Indeed, the transitional period from spring to neaps (a period when the bank is still affected by

high degrees of saturation due to the high spring tides, but the confinement pressure favoured by the elevated water level is declining) creates conditions when failures are more likely.

Although these controlling factors are very similar to those that have been documented previously in fluvial environments (where prior studies have emphasised the destabilising conditions encountered on the falling limbs of flood hydrographs), a crucial difference is the higher inundation frequency that is characteristic of tidal environments. Fluvial settings are

characterised by less repetitive high-low water alternations and normally are governed by changes in the hydrograph due to seasonal (e.g. monsoon areas) rainfall patterns. These longer timescales leave enough time for a balancing of the groundwater



table (and the associated pore pressures) with the wetting front of the rising/dropping water level. Markedly, in riverine settings, mass wasting events have been shown to be favoured by the presence of complex hydrographs where a series of minor precursor peaks precede the main high water event (e.g. Luppi et al., 2009; Rinaldi et al., 2008, 2004). Only in such conditions

(similar to what occurs in the alternation of spring-neap conditions in tidal settings) is a river bank subject to repetitive, nearly saturated, conditions that can generate the positive pore water pressures needed for a rise in instability and a series of failures.

In contrast, in tide-dominated (hypertidal) cohesive banks, the very regular and rapid alternation between high-water and low-water stages plays a central role in controlling the interplay between stabilising (cyclical high water levels leading to regular high confining pressures) and destabilising (high levels of pore pressure) forces. The onset of unstable conditions (FoS<1) in

tidal fine-grained settings occurs quickly (in ~10 minutes in the studied area) after the hydrostatic confining pressure drops below the bank top and while the internal pore water pressure is still high. Under these circumstances, when the hydrostatic confinement pressure is absent (low tides), the bank experiences a rapid drop in the factor of safety. Thus, in tidal environments and in particular in hypertidal settings where the tidal range is very large and the overall range of hydrostatic confinement pressures is likewise very large, because of the regular alternation of high and low tidal water phases, both during a single day

and (more markedly) during the spring-neap periodic cycle, the likelihood of mass wasting events increases.

A synthesis of the dominant factors influencing bank stability in tidal and fluvial settings is presented in Figure 11, which is subdivided into tidal (left-hand panels) and fluvial (right-hand panels) domains. For the tidal domain, the water level (indicated by the blue dot in the diagrams) follows a path that is identical to that for Scenario 1, starting from a high water level (similar to phase P1 in Fig. 9b and c), while for the fluvial domain (indicated by the black dot in the diagrams) the water level is based

on typical hydrographs published in prior studies of rivers (e.g. Casagli et al., 1999; Luppi et al., 2009; Rinaldi et al., 2004) and also commences at a high flow stage.

In the hypertidal regime, during periods of high spring tides that completely inundate the bank (Fig. 11a), the two driving forces (destabilising pore pressure and stabilising confinement pressure) are both at their maximum; on the one hand, the bank material is fully or nearly fully saturated (with an extended development of positive pore water pressures resulting in a

reduction in the effective stress) which promotes instability; on the other hand, the stabilising confining pressure of the tidal level is exerted along the full extent of the bank profile. This stabilising influence is greater in magnitude than the destabilising influence of the elevated pore pressure so the bank remains in a stable condition (high levels of FoS). As the tidal waters begin to recede (phase P2 and point B in the panels on Fig. 9) and during the continued ebbing phase (Fig. 11b and c), the confining pressure is gradually decreasing, and the onset of the drawdown process triggers a transient seepage component in the direction

of the channel. However, the low hydraulic transmissivity of the estuarine fine-grained deposits does not allow for complete de-saturation of the soil; hence, during these phases, the bank experiences a condition in which the pore water pressure ($u$) destabilising effect remains high, but the stabilising confining pressure is markedly reduced (Fig. 11c). At this point, and specifically when the water level falls below the incipient failure surface threshold elevation of 6.4 m at which the confining



pressure falls to zero (for the specific case of the study site), the imbalance between the destabilising pore water pressure and

stabilising confining pressure is maximised and the FoS therefore drops swiftly (Fig. 11d). As demonstrated in this study, the

drawdown phase after an overbank flow condition is also often followed by intense drainage from the floodplain. Such drainage

events likely also favour hydraulic erosion (as exemplified by the drone images), independent of the mass-wasting dynamics,

that would further compound any inherent instability.

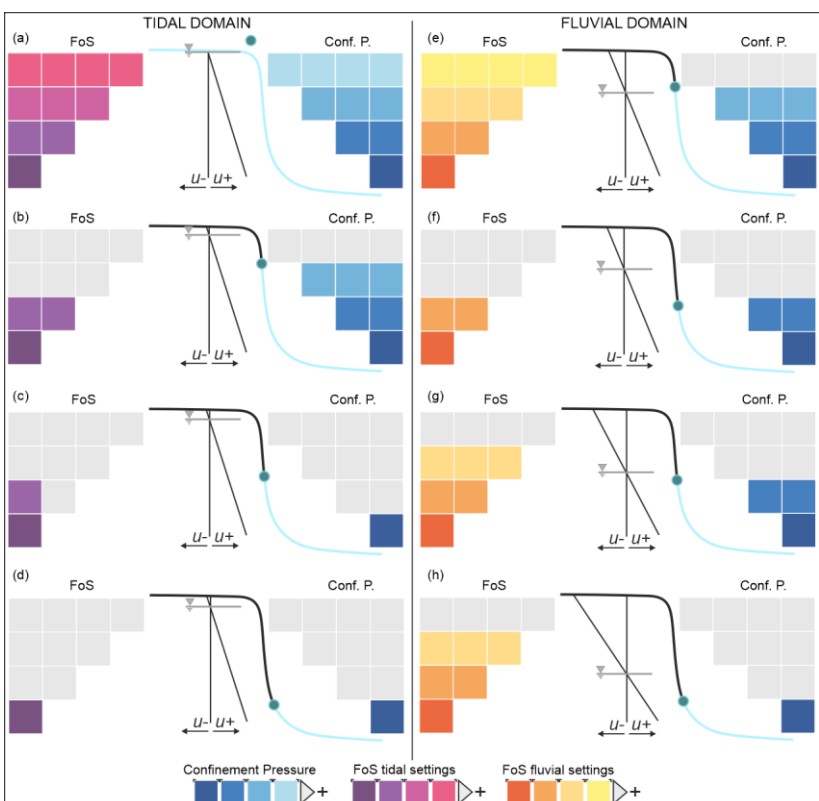

**Figure 11. Conceptual model of the factors controlling the onset of bank instability in fine grained tide-dominated hypertidal settings (such as the Severn estuary) versus fluvial-dominated environments.**

In both tidal and fluvial settings, the relationship between the evolution of positive pore pressures, the hydrostatic confining

pressure, and factor of safety is somewhat similar with a general increase in stability conditions during high water periods,

followed by destabilisation as a result of the emergence of elevated pore pressures coinciding with a decrease in hydrostatic

confinement during drawdown phases. However, differences between tidal and fluvial settings are apparent in the timing of

the alternation between more stable and unstable periods. Within fluvial settings (Fig. 11 fluvial domain), these stability –

instability switches are often spread over many hours (days), permitting the bank to equilibrate its groundwater table (and the

pore pressures) with the wetting front induced by the surging – dropping water level (Fig. 11f and g). This lag effect is

particularly evident in systems with pronounced seasonal differences in water stage. Furthermore, in fluvial systems, it is less

common to find uniform and very fine-grained banks, and a fully saturated bank body (in general, the water stage in a river is



below the floodplain level for the majority of the year). These two conditions are less frequently met in fine-grained (and therefore low hydraulic transmissivity) hypertidal settings like the Severn Estuary. In such tidal settings, the alternation of very high water stages and low water stages occurs repeatedly over timescales of just a few hours. The bank is therefore subject to a continuous state of near saturation without the possibility of establishing a lasting seepage outflow, meaning that pore water

pressures remain more elevated during the recession period than in equivalent fluvial settings.

## 5 Conclusions

The present study combines high-resolution monitoring and modelling at the Plusterwine study site in the middle Severn Estuary (UK) to elucidate the detailed factors driving observed mass wasting events in a hypertidal cohesive-banked estuary. The results show how the conditions that lead to the onset of bank failure in hypertidal settings depend on a variety of factors

such as the bank geometry and the properties of the bank materials, and (critically) the relationship between pore water pressure destabilising trends and the stabilising effect afforded by the hydrostatic channel confining pressures, both of which are linked to the tidal forcing. It is evident that, regardless of the considered environment (tidal versus fluvial), the occurrence of the drawdown stage (whether during the tidal ebb or the recession limb of a fluvial hydrograph) is a crucial element in controlling the bank stability. For the typically finer materials of tidal environments, the bank deposits remain in a near-saturated state at

precisely the point in time when the hydrostatic confining pressure is decreasing, paving the way for the pore pressures to destabilise the bank and favour the onset of mass failures. This mutual role of stabilising and subverting forces leading to bank failure is common to both river and tidal environments; however, the higher frequency (semidiurnal) and cyclical (spring-neap) alternations of the high and low water stages (very high and very low during spring tides) seems to play a fundamental role in increasing bank instability with respect to mass failure in tidal settings. If on the one hand the lower inundation

frequency characteristic of riverine settings favours the presence of just a few moments (few hours) during a typical year in which the river bank is subject to conditions of instability, on the contrary, the recurrent (semidiurnal) presence of high/low water levels characteristic of tidal environments enhances the onset of multiple opportunities for instability: (i) fast drainage flows from the top of the tidal flat (gullies), (ii) persistence of saturated conditions, and (iii) periodic removal of hydrostatic confining pressures (low tidal conditions).

**Acknowledgements**

This work was completed as part of Andrea Gasparotto's PhD studentship, which was funded by the UK Research and Innovation (UKRI) program. Special thanks to the Environment Agency and Natural England for the support offered in accessing the study site and to the local landowners and tenants for the assistance during fieldwork activities.



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
