# Peer review of "Water level fluctuations drive bank instability in a hypertidal estuary"

_Earth Surface Dynamics, 2022_

## Author Comment (AC1)

esurf-2022-44. *Water level fluctuations drive bank instability in a hypertidal estuary*.

[Figure]

Figure 2 amended as per RC1.

---

## Author Response (AR1)

**REFEREE #1 comments and answers:**

The paper concentrates on bank erosion processes in hypertidal estuary environment. The research has been well justified, and the results and discussion are very valuable to scientific community within the field of geomorphology, coastal processes and related fields. The paper has been written clearly and structure is also very good. I have only minor comments.

- **Comment 1)** Introduction: the last paragraph is heavy and I suggest that it would be split in couple of paragraphs. For example lines 51-55 could be a separate paragraph.

  **Response:** Thank you for the suggestion. We have simplified the text, shortening it a little, while also breaking the original paragraph down into two more accessible ones in the revised manuscript. Here below the revised introduction (including the tracked changes):

  … The evolution of fine-grained shorelines within estuaries is closely connected to bank retreat processes (Zhang et al., 2004, 2021; Guo et al., 2021; Zhao et al., 2022).  While beach retreat (Jolivet et al., 2019; Bain et al., 2016; Hird et al., 2021; Carvalho and Woodroffe, 2021; Masselink et al., 2016) and cliff erosion (Brooks et al., 2012; Leyland and Darby, 2008; del Río and Gracia, 2009; Young et al., 2014; Hackney et al., 2013) have been well researched, sensitive estuarine environments have received less attention despite their societal importance. About 60 % of the world's population is concentrated along coasts, and 22 of the largest cities on Earth are located adjacent to  estuaries (Harris et al., 2016). Furthermore, estuarine environments such as salt marshes  are essential in the mitigation of coastal flooding, attenuating  wave activity (Möller et al., 2014; Fairchild et al., 2021; Leonardi and Fagherazzi, 2015), and aid carbon sequestration  (Li et al., 2022; Pendleton et al., 2012).

  Although some  studies (Bendoni et al., 2014; Mel et al., 2022; Carniello et al., 2009; Marani et al., 2011; D'Alpaos et al., 2007) have explored marsh retreat behaviours in microtidal settings (e.g. Venice Lagoon, Italy), and others (Shimozono et al., 2019; Roy et al., 2021) have investigated  erosion in large tidal-dominated estuaries , studies that consider the problem of bank collapse geomechanically, and with a particular focus on hypertidal environments, are lacking. Given the centrality of estuaries as transitional zones between the sea and land, a more complete understanding of the sources, mechanics and rates of bank erosion  is of substantial importance. Yet, b failure processes  in tidal settings have to date been poorly studied and quantified (Gong et al., 2018; Zhao et al., 2022, 2019), especially when compared with the large literature on bank erosion in non-tidal (fluvial) environments (e.g. Rinaldi and Nardi, 2013; Nardi et al., 2012; Patsinghasanee et al., 2018; Julian and Torres, 2006; Darby and Thorne, 1996b; Darby et al., 2000; Majumdar and Mandal, 2022; Zhang et al., 2021; Thorne and Abt, 1993; Darby et al., 2010; Duong Thi and do Minh, 2019). Given the additional complexity of the process mechanics involved in tidal settings, arising mainly from the presence of bidirectional flows, process insights gained from studies of fluvial bank erosion may not necessarily be transferable to estuarine contexts. The present study seeks to address this gap through an investigation in which a combination of field observations and geotechnical modelling is employed to elucidate the bank failure processes operating in a hypertidal environment (the Severn estuary, UK).

- **Comment 2)** Line 53: It could be good to mention related to the field observations, how many years and when the measurements took place?

  **Response:** We propose to amend the text as follow (toward the end of section 1):

  The present study seeks to address this gap through an investigation in which a combination of field observations (made during the period 2018 to 2020; for details the reader is referred to Table 1) and geotechnical modelling is employed to elucidate the bank failure processes operating in a hypertidal environment (the Severn estuary, UK).

- **Comment 3)** Lines 115-116: Did the drone also have accurate (RTK-GPS) location information, or was the georeferencing solely based on the GCPs?

  **Response:** We clarify that the drone does not have its own georeferencing system; georeferencing was accomplished by using Ground Control Points (GCPs) linked to an accurate grid of topographic surveys. We don't think we need to modify the manuscript because the application of the GCPs in the georeferencing process is already discussed in the manuscript (section 2.2.1 Aerial surveys).

- **Comment 4)** Figure 2: Where does the P-3 picture locate on the left hand side's map? I cannot find it from the map, only P-1 and P-2 are shown. In photos P-1, P-2 and P-3 the direction of the view is different to each other. Could it be possible to show for example the north direction in these photos, so that the reader could easier interpret the figure and which part of the bank is seen in each photo.

**Response:** We are pleased to clarify that P-3 is located roughly at the centre of the image on the left-hand side of the figure. In the original manuscript the boxes representing the snapshot locations are slightly too dark and therefore in the revised paper we have revised them to better highlight the locations and we have also added north directions as per the reviewer's helpful suggestion.

[Figure]

How Figure 2 was

New version of Figure 2 (north arrows on lateral images and location of the three snapshots in red boxes.

- **Comment 5)** Line 196: In which laboratory the triaxial shear tests were performed? Could you please explain a little bit more about the triaxial shear test procedure. How large was the grain size, and what kind of approach/equipment were used for the tests?

**Response:** We propose to add the following text in section 2.3:

The triaxial tests were performed in the Geomechanics Laboratory at the Civil, Maritime and Environmental Engineering department of the University of Southampton, following the British Standard methods (British Standard Institution, 1990). All the tests were conducted using an electro-mechanical TRITECH machine with a maximum compression capacity of 10 kN. The gain size distribution of the two analysed layers is reported in Table 3. The results presented here provide insight into the detailed processes leading to the observed mass wasting events.

- **Comment 6)** Line 200: "preliminary tests" -> what tests are meant with this, does it refer to triaxial shear tests or model tests? This sentence could be clarified.

  **Response:** We have revised the paper to clarify that this refers to additional tests (not triaxial tests) to acquire the additional parameters used in the model setup. We have modified the text to read (section 2.3):

  Prior to undertaking triaxial tests, a portion (c. 40 mm diameter and 80 mm length) of the soil samples retrieved from the field site was removed by cutting the sample in order to estimate the moisture content and unit weight of the material. Moisture content was obtained by using the oven-drying method as stated by the British Standard (British Standard Institution, 1990) for geotechnical laboratories. First, the wet mass of the sample was measured before placing it into an oven for 24 hours at 105°C. After 24 hours the dry sample was weighed again and the moisture content (%) calculated. The unit weight of each sample was then calculated by dividing the sample mass by its volume.

- **Comment 7)** Table 3: How was the friction angle and water contents derived? It seems that these were not described in the methods section yet. Could you please clarify, and add shortly about these calculations?

  **Response:** We have revised the text to clarify these points. Specifically, we now include the following text (added at the bottom of page 11) to address the method clarification:

  After the triaxial testing, the saturated water content was calculated for the different samples. At the end of the triaxial testing procedure the samples were fully saturated, so that the saturated water content was estimated using:

  $\theta = (m_2 - m_3) / (m_3 - m_1)\ 100$

  where $m_1$ is the mass of the sample container, $m_2$ is the is the wet soil mass plus container, and $m_3$ is the dry soil mass plus container. Dry soil mass was determined before the triaxial tests on undistributed samples collected from the bank blocks.

- **Comment 8)** Line 231: How it was decided that the resolution will be 0.5 m? Or do "sensitivity tests" on lines 251-252 refer to these test of grid size impact on results? Please, could you add in the methods section more clearly how the resolution was selected to be 0.5 m. These parts of methodology section need clarification.

  **Response:** We are pleased to confirm that the reviewer is correct: the sensitivity tests allowed the definition of the most appropriate grid resolution (0.5 m). Specifically, comparisons between a coarser and a more refined mesh indicated that model results are insensitive to the selected grid design selected. Thus, a grid resolution of 0.5 m is appropriate to resolve the key processes of concern here.

  We suggest a modification in the text to read as follow (page 13 – revised manuscript):

  Note that a series of sensitivity analyses were carried out to ensure the robustness of the model setup process. Specifically, these sensitivity tests were designed to demonstrate that the simulations are independent of variations in the discretisation of the selected finite element grid, as well as of variations in those model boundary conditions that were specified based on estimated values rather than measurements. These sensitivity tests allowed the definition of the most appropriate grid resolution and, markedly, the comparison between coarser and a more refined mesh indicated that the model results are insensitive to the selected grid design. Therefore, a grid resolution of 0.5 m was considered appropriate to resolve the key processes occurring in the studied bank. Regarding the model boundary conditions, these  refer specifically to the zero flux conditions assigned to the left lateral and basal horizontal boundaries. The sensitivity tests revealed that the simulated pore water pressures within the materials close to the bank face, which are subject to the investigated bank collapses, are independent of the assumed zero flux conditions. Similarly, comparisons between a coarser and a more refined mesh indicate that the model results are insensitive to the discretised grid design selected here.

**REFEREE #2 comments and answers:**

This study addresses the bank failure problem at hyper-tidal estuary, which is well written and fairly presented. I have only a minor concern and comment: the tidal level at the study site can be predicted rather accurate by various models, it should have been better if tidal level at the site was used to drive the model. Particularly for the long term simulation, the impact may be accumulated over time.

**Response:** Thank you for the comment. We are fully aware of the variation in the tidal level between the study-site and the used station. However, we have applied a correction to all the data utilised in the model to account for this difference in the distance between the station and the study area (5 km distance) as reported in line 384-393 (original preprint). We are confident that the model and the subsequent analyses well-represent the conditions at the study-site and, particularly, that the tidal levels used in the model and corrected using the time-delay appreciated with the on-site monitoring are representative of the Plusterwine area tidal conditions.